# Frailty Prevalence and Association with Clinical Outcomes in Interstitial Lung Disease, Asthma, and Pleural Disease

**DOI:** 10.3390/geriatrics8040082

**Published:** 2023-08-13

**Authors:** Alessia Verduri, Ben Carter, Ceara Rice, James Laraman, Eleanor Barton, Enrico Clini, Nick A. Maskell, Jonathan Hewitt

**Affiliations:** 1Department of Population Medicine, Cardiff University, Cardiff CF10 3XQ, UK; alessia.verduri@unimore.it (A.V.); ceara.rice@wales.nhs.uk (C.R.); james.laraman@wales.nhs.uk (J.L.); 2Respiratory Unit, Department of Surgical and Medical Sciences, University of Modena and Reggio Emilia, 41124 Modena, Italy; enrico.clini@unimore.it; 3Department of Biostatistics and Health Informatics, Institute of Psychiatry, Psychology and Neuroscience, King’s College London, London SE5 8AF, UK; ben.carter@kcl.ac.uk; 4Academic Respiratory Unit, School of Clinical Sciences, University of Bristol, Bristol BS10 5NB, UK; eleanor.barton@nbt.nhs.uk (E.B.); nick.maskell@bristol.ac.uk (N.A.M.)

**Keywords:** interstitial lung disease, Idiopathic Pulmonary Fibrosis, asthma, pleural disease, frailty, prevalence, mortality, hospitalisations

## Abstract

Background: Frailty is a syndrome characterised by increased vulnerability to negative outcomes. Interstitial lung disease (ILD), asthma, and pleural disease are leading causes of morbidity and mortality. We aimed to investigate the prevalence and impact of frailty in adult patients with these diseases. Methods: We conducted a systematic review and meta-analysis, searching PubMed, Web of Science, The Cochrane Library, and EMBASE for studies reporting on frailty in ILD, asthma, and pleural disease. MeSH terms including interstitial lung disease, Idiopathic Pulmonary Fibrosis, Non-specific Interstitial Pneumonia, Chronic Hypersensitivity Pneumonitis, systemic sclerosis-associated ILD, connective tissue disease-associated ILD, and frailty were used as key words. The primary outcome was prevalence of frailty. Where enough contextually homogeneous studies were included, a pooled random-effects meta-analysis was performed with mortality and hospitalisation as the outcomes. Results: The review found three studies relating to frailty in asthma. No studies relating to pleural disease and frailty were identified. The median prevalence in asthma was 9.5% (IQR, 7.8–11.3). Six relevant studies incorporating 1471 ILD patients (age 68.3 ± SD2.38; 50% male) were identified, which were either cohort or cross-sectional design rated either good or fair. The median prevalence of frailty was 48% (IQR, 25–50). There was a positive association between frail ILD patients and increased risk of long-term mortality (pooled OR, 2.33 95%CI 1.31–4.15, *I*^2^ 9%). One study reported a hospitalization rate of HR = 1.97(1.32–3.06) within 6 months in frail ILD patients. Conclusions: Frailty is very common and associated with increased mortality in patients with ILD. There are still minimal data regarding the prevalence of frailty and its influence on the risk in this population.

## 1. Introduction

Frailty is a syndrome characterised by a decline in physiological reserves and functions resulting in a greater vulnerability to negative health outcomes following acute stressors [1,2]. Chronic diseases, including lung diseases, are known risk factors for the development of frailty [1,2,3].

The prevalence of frailty varies according to the criteria or model used to identify it and the setting in which a population is studied [4]. Its prevalence increases with age, although it is also seen in younger populations, especially those affected by chronic conditions [5,6,7]. Commonly, frailty is measured using a Phenotype model [2] or by an accumulation of deficits [8]. These models have been developed according to a different underlying concept of frailty. The frailty index is a quantification of the cumulative burden of health deficits and widely based on chronic diseases. Conversely, the Phenotype model is based on domains such as weakness, slowness, and low physical activity. This model includes an assessment of physical function that, when impaired, can cause disability independently of diseases. Some of the clinical manifestations of frailty are due to sarcopenia that is directly related to low physical performance [9].

Prevalence of frailty may vary according to the model used (deficit or phenotype) and prevalence rates vary according to the setting, for example in a younger community-based population, such as the UK Biobank, it was as low as 3% [10] but in a large hospital population with COVID-19, with an average age of 74 years, the prevalence was 49.4% [7]. Increased frailty has been shown to consistently correlate with worse outcomes across a wide range of medical specialities and populations [11]. If chronic diseases are considered the cause of frailty, it becomes extremely important to assess frailty in patients with long-term conditions. More personalised medicine should be beneficial following evaluation of the role of frailty in disease course and intensive treatments [12].

The term interstitial lung disease (ILD) includes a heterogeneous group of diffuse parenchymal pulmonary disorders, with varying degrees of inflammation or fibrosis [13,14]. The literature on frailty in ILD has been evolving in recent years but there remains a paucity of knowledge in the area of frailty, despite frailty being a common risk factor for higher morbidity and early mortality in patients with ILD, independent of age [15].

Asthma is an important global health problem affecting millions of people and all age groups, with a rising burden on health care systems and on society through loss of productivity in the workplace [16,17]. The evidence for the effect of frailty in patients with asthma has not been explored to any significant extent.

Pleural disease remains common, affecting 360 per 100,000 people each year [18]. Pleural disease is the result of a range of different pathologies and includes malignancy, organ failure, infection, and pneumothorax. Collectively, it represents a significant burden in clinical practice [18].

The literature of frailty in Chronic Obstructive Pulmonary Disease (COPD) is evolving. This topic was the subject of another systematic review [Verduri A et al. Manuscript under review], therefore COPD has been excluded in the present review.

The aim of this study was to perform a systematic review and meta-analysis of observational studies to explore the role of frailty in adult patients with ILD, asthma, and pleural disease. The specific aims were (1) to identify the prevalence of frailty in people with ILD, asthma, or pleural disease; and (2) to assess the association between frailty and mortality, between frailty and morbidity such as exacerbation rates, and the number of hospitalisations.

## 2. Materials and Methods

The systematic review and meta-analysis were conducted in accordance with the Preferred Reporting Items for Systematic Reviews and Metanalyses (PRISMA) recommendations. The protocol was registered through the PROSPERO database (registration number: CRD42022328511).

### 2.1. Search Strategy and Study Selection

The search strategy was developed with a specialist librarian. Two researchers (AV, CR) independently searched four electronic databases (PubMed, Web of Science, The Cochrane Library, and EMBASE) for manuscripts published from inception to 25 October 2022. The search terms were based on Medical Subject Headings (MeSH). MeSH terms referring to frailty and Interstitial Lung Disease were used as key words, including the following: Idiopathic Pulmonary Fibrosis, Non-specific Interstitial Pneumonia, Chronic Hypersensitivity Pneumonitis, systemic sclerosis-associated ILD, connective tissue disease-associated ILD, and frailty. MeSH terms referring to frailty and asthma were used as key words. The review process for ILD is summarised in a PRISMA flow diagram (Figure 1). The search strategy is outlined in Appendix A. The PRISMA flow diagrams for asthma and pleural disease are shown in Appendix A. Studies reporting information on frailty assessment and interstitial lung disease, asthma, and pleural disease in titles and abstracts were included. Any disagreement on study eligibility was resolved through discussion with a third reviewer (JH). A hand search of the reference list of all relevant articles was performed to identify any articles not captured by the electronic search. Only full-text reports in English and Italian were considered.

The Inclusion criteria were (1) study participants (≥18 years) who were diagnosed with ILD, asthma, or pleural disease who were assessed for frailty; (2) only studies using a validated method of frailty identification were included; (3) primary outcome measure was the prevalence of frailty in adults with ILD, asthma, or pleural disease; (4) cross-sectional, longitudinal, prospective or retrospective cohort, and case–control study designs.

Articles based solely on editorial, commentary, symposium, expert opinion, congress abstract, ongoing studies not published, short reports, and reviews were excluded.

Exclusion criteria were ILD patients listed for or referred for lung transplantation.

### 2.2. Data Extraction and Quality Assessment

Study characteristics, demographic information, frailty tool, frailty prevalence, and outcomes data were independently extracted from the included studies. Study authors were contacted to clarify or provide additional data where it was missing or unclear.

For included studies, the quality assessment was conducted by the two reviewers independently (AV, CR) and arbitrated by a third (JH) using the Newcastle–Ottawa Scale (NOS), [19] which assesses the risk of bias of observational studies. Each domain examined was classified as good, fair, or poor corresponding to low, moderate, and high risk of bias, respectively. Studies were considered to be of good quality where they scored good for all domains, fair if they scored fair in one or more domain, and poor if they scored poorly in any domain.

### 2.3. Outcomes

The primary outcome was prevalence of frailty in ILD, asthma, and pleural disease. Secondary outcomes were mortality; exacerbation rate; or number of hospitalizations in patients with ILD, asthma, or pleural disease. A priori, long-term mortality was defined as ≥1-year mortality.

### 2.4. Key Exposure of Frailty

Studies were included if they measured frailty using any validated instrument [20]. The following were examples of those included: Fried Frailty Phenotype [2], Frailty Index [8,21], and Kihon Checklist [22]. Frailty was measured as a binary variable as frail or not frail using the thresholds presented for the individual instruments. Where frailty measures included pre-frailty, this variable was also included. For example, in the Fried Frailty Phenotype, pre-frailty is defined as the presence of one or two of the following factors: unintentional weight loss, self-reported exhaustion, slowness, weakness, and low physical activity [2]. We summarized each frailty instrument in Appendix A.

### 2.5. Data Analysis

The primary outcome of the prevalence of frailty was estimated as the median study level prevalence, presented alongside the range.

Secondary outcomes were narratively described, and where study characteristics were deemed as contextually homogeneous, they were associated with frailty as a binary variable. Data were analysed in a pooled random-effects meta-analysis using the Mantel-Haenszel method. Pooled estimates were presented as OR with the associated 95% CIs, *p* values, and *I*^2^ summary data. RevMan V5.4. was used as statistical software.

### 2.6. Assessment of Subgroups and Statistical Heterogeneity

Statistical heterogeneity was measured using the *I*^2^ statistic. Heterogeneity exceeding 80% was explored using subgroup analyses. Pre-specified subgroups to explore heterogeneity included quality assessment.

## 3. Results

### 3.1. Interstitial Lung Disease

#### 3.1.1. Search Results and Quality Assessment on ILD Studies

After removal of duplicates, 135 records were identified. Sixty-seven (67) full texts were reviewed, and 61 of these were excluded with reasons. The six studies included in the analysis are shown in the PRISMA flowchart (Figure 1). Three studies were determined as good quality [23,24,25] and three were categorized as fair quality [26,27,28]. The qualitative synthesis is summarised in Appendix A.

#### 3.1.2. Characteristics of Studies on ILD Included

A summary of the characteristics of included studies is presented in Table 1. The included studies were published between 2017 and 2022. Included studies were conducted in Canada (*n* = 5) [23,24,25,27,28] and the USA (*n* = 1) [26]. From the six studies, 1471 patients were included, 50% of which were male (735/1471). There was a range of frailty assessment tools used in the included studies, of which all were deemed suitable for inclusion in the frailty prevalence estimation. The mean age was 68.3 (±SD 2.38) and the age ranged between 48 and 79 years (Table 1).

#### 3.1.3. Frailty Prevalence in ILD

Prevalence was assessed using two different frailty scales: the Fried Frailty Phenotype [2] and the Frailty Index (Appendix A) [8,21]. Of the six included studies, the median prevalence of frailty in ILD patients was 48% (25–50 IQR), ranging from 25% [25] to 52.5% [27]. The presence of pre-frailty ranged between 20% [27] and 56%, [23] with a median of 40% (IQR, 23–55). The included studies on ILD and the prevalence estimate are shown in Table 1.

#### 3.1.4. Frailty Score as a Predictor of Morbidity and Mortality in Patients with ILD

##### Long-Term Mortality

Two studies [23,24] explored frailty, pre-frailty, and mortality in the long term, including 1003 patients. Both studies found a positive association over a median time ranging between 17 months and 3 years. The pooled analysis showed a positive association between ILD patients living with frailty and increased risk of long-term mortality compared to non-frail ILD patients (pooled OR, 2.33 95% CI 1.31–4.15, *I*^2^ 9%) (Figure 2). When frailty and pre-frailty were compared against non-frailty and frailty was compared against pre- and non-frailty, the results showed a trend towards worsening mortality for the frailer people (Appendix A).

Due to the low number of included studies on ILDs, no sensitivity analyses were carried out.

##### Number of Hospitalizations

One study [24] reported a nearly double rate of all-cause hospitalisations (HR = 1.97, 1.32–3.06, *p* = 0.002) within 6 months after frailty assessment in ILD patients living with frailty.

### 3.2. Asthma

#### 3.2.1. Search Results and Quality Assessment on Asthma Studies

We identified three studies regarding frailty in asthma (Table 2; Appendix A) that were reviewed and quality assessed. Two studies were categorized as fair quality, [29,30] and one study as good quality (Appendix A) [10]. Due to the low numbers of included studies, meta-analyses were not performed.

#### 3.2.2. Characteristics of Included Studies on Asthma

A summary of the characteristics of included studies is presented in Table 2. The included studies were published between 2018 and 2021. Included studies were conducted in France, [29] Japan, [30] and the UK (Table 2) [10]. From the three studies, 57,610 participants with asthma were included. The mean age was 69.6 and the age ranged between 37 and 80 years. Data from the UK Biobank [10] did not describe prevalence of sex and age range in the group of participants with asthma. There were two frailty assessment tools (Fried Frailty Phenotype and Kihon Checklist) [2,22] used in the included studies (Appendix A), and all studies were deemed suitable for inclusion in the frailty prevalence estimation. The three studies identified did not examine the impact of frailty on clinical outcomes in asthma.

#### 3.2.3. Frailty Prevalence in Asthma

Of the included studies, all [10,29,30] reported the prevalence of frailty in patients with asthma and the median prevalence was 9.5% (IQR, 7.8–11.3), ranging from 6% [10] to 14.5% [30]. The overall prevalence of pre-frailty ranged between 30.4% [30] and 42.1% [10], with a median of 38% (IQR, 30.4–42.1). The included studies on asthma and the prevalence estimate are shown in Table 2.

### 3.3. Pleural Disease

#### Search Results on Pleural Disease Studies

We did not find any studies relating to frailty in pleural disease.

## 4. Discussion

### 4.1. ILD

This study is the first review that systematically characterises frailty in ILD. The results showed that the prevalence of frailty or pre-frailty was present in nearly 50% of the study participants and that there was an association between frailty and increased mortality.

The study identified six studies on ILD, which included 1471 patients [23,24,25,26,27,28]. Three studies were good quality [23,24,25], and three were fair quality [26,27,28]. The included studies covered a range of interstitial lung diseases, including Idiopathic Pulmonary Fibrosis (IPF), Chronic Hypersensitivity Pneumonitis, systemic sclerosis-associated ILD, Non-specific Interstitial Pneumonia, connective tissue disease-associated ILD (CTD-ILD), and other unclassifiable ILDs.

Frailty was found in 50% of patients with ILD aged 60 years and over, confirming that age alone represents a narrow field of view in patient management and showing a potential relationship between interstitial lung disease and ageing. The high proportion of ILD outpatients living with frailty and assessed in stable condition is likely related to a degree of physical inactivity that widely contributes to sarcopenia and progressively to more severe frailty [9]. ILD patients living with frailty can have higher vulnerability and dramatically be affected by even small acute stressors that shorten their survival [15].

Two prospective cohort studies [23,24] reported a positive association between frailty and long-term mortality in patients with ILD in stable conditions, with a more than double chance of dying in frail people. Similarly, other lung diseases such as Chronic Obstructive Pulmonary Disease (COPD) showed an increased risk of mortality in patients with COPD living with frailty [Verduri A et al. Manuscript under review]. The type of ILD patients varied between the studies. Guler et al., [24] included 18.5% patients with IPF and 42% patients with SSc-ILD and other connective-tissue disease ILD, and the study of Farooqi et al. [23] involved 39.5% patients with IPF and 17% patients with CTD-ILD. These different patient groups may have introduced some bias. However, it is well known that systemic sclerosis and IPF are associated with other age-related comorbidities that can facilitate the development of frailty. Both studies considered the comorbidities associated with ILD. Farooqi et al. [23] did not find any difference in the number of comorbidities between frail and non-frail patients, while Guler et al. [24] suggested that the co-occurring diseases influence the frailty score in non-connective tissue disease-ILD. Further investigations on frailty measurement including comorbidities in ILD are required.

Only one study [24] reported all-cause hospitalisations, with a nearly double rate in ILD patients living with frailty within 6 months.

While these findings confirm the role of frailty in worse clinical outcomes of patients with ILD, other studies on frailty and a broader range of relevant clinical outcomes are required.

A strength of this systematic review is that it included the Fried Frailty model [2] and the accumulation of deficits model (Frailty Index) [8,21] that are the two of the most commonly employed frailty instruments, both valid and frequently applied to research and used in clinical practice. The populations considered were North American, which may limit generalisation and the comparatively low numbers of studies did not allow meta-analyses across a wide range of outcomes and this must be noted as a limitation. Further, these data are observational and reverse causality is possible, if unlikely.

Increasing evidence has emerged on progressive fibrosis in patients with IPF and also with different underlying interstitial lung disease. The progressive-fibrosing phenotype can be characterised by worsening respiratory symptoms and early mortality [14]. Prevalence of frailty may vary in fibrotic ILD and non-fibrotic ILD. Whilst this can be a potential limitation of the study, it also suggests that future studies on frailty and clinical outcomes in ILD should take the phenotype into account.

Recently, Saketkoo et al. [31] focused on patients with ILD in advanced age and in people with late-in-life onset ILD, proposing prevention of frailty in ILD at any age for well-being of this group of patients.

### 4.2. Asthma

The review identified only three studies on asthma and frailty, which involved 57,610 individuals [10,29,30]. Whilst older age was prevalent in two studies [29,30], one study [10], that included 57,169 people with asthma, reported 81% of all the individuals recruited were aged <65 years. All studies were found to be of fair quality, [29,30] except one of good quality [10]. The studies included in this review showed a prevalence of frailty in asthma of 9.5% and pre-frailty of 38%.

Although all included studies were reliable and used validated frailty tools, one study [30] assessed outpatients with diagnosis of asthma according to the GINA guidelines [17] and the two other studies [10,29] through questionnaires, meaning that the majority of participants had a mostly self-reported asthma diagnosis. Further, there was little specific evidence regarding the severity of the asthma and the disease in older populations. Particularly, in older patients, the asthma prevalence does not decrease with ageing and the disease has specific clinical presentation in advanced age, such as asthma-COPD overlap syndrome as an example. The recognition of asthma in ageing is frequently delayed and the treatment postponed due to difficulties that clinicians have in the approach in older asthmatic patient. Recently, a study protocol on the role of frailty in older patients with diagnosis of moderate-to-severe asthma according to guidelines [17] has been published by a group of researchers from Brazil, which may help characterise this group of patients [32]. A greater number of future studies and a multidisciplinary management of asthma in older adults are still needed.

### 4.3. Pleural Disease

This study did not find any articles regarding pleural disease and frailty. The measurement of frailty in patients with pleural disease has not been considered until now, this is therefore a clear gap in clinical practice and research. 

## 5. Conclusions

This work included a limited number of studies on ILD but those that are reported provided consistent and plausible findings. The study found that frailty is common in patients with ILD. Although frailty has been recently recognised as an important aspect for patients with ILD, there is still paucity of data regarding the prevalence of frailty and mainly its influence on clinical outcomes in this population. Despite only two studies included in the meta-analysis, this review showed that frailty was associated with increased long-term mortality in this population. However, additional studies are required to determine whether frailty can be used to predict disease progression and worse health outcomes. These studies will clarify how the frailty status can be reversed, and potentially identify therapeutic targets for improved outcomes. Dyspnea, cough, and fatigue are frequent in ILD before dying and contribute to physical inactivity, deconditioning, and further worsening of symptoms that can lead to high degree of frailty in these patients. This systematic review outlines the challenges facing ILD clinicians in making an early identification of frailty with resulting non-pharmacological approaches in the care of patients.

In conclusion, the study provided a scientific summary of the currently available literature regarding frailty and ILD, asthma, and pleural disease and identified gaps that can be used for future research. Frailty is evolving as an important measure in chronic lung disease such that may enhance the clinical management of respiratory patients at risk. Areas such as severe asthma, asthma in older populations, and pleural disease have not been well described or considered and require further investigation. These future directions of research and clinical practice on frailty could potentially improve patient outcomes.

## Figures and Tables

**Figure 1 geriatrics-08-00082-f001:**
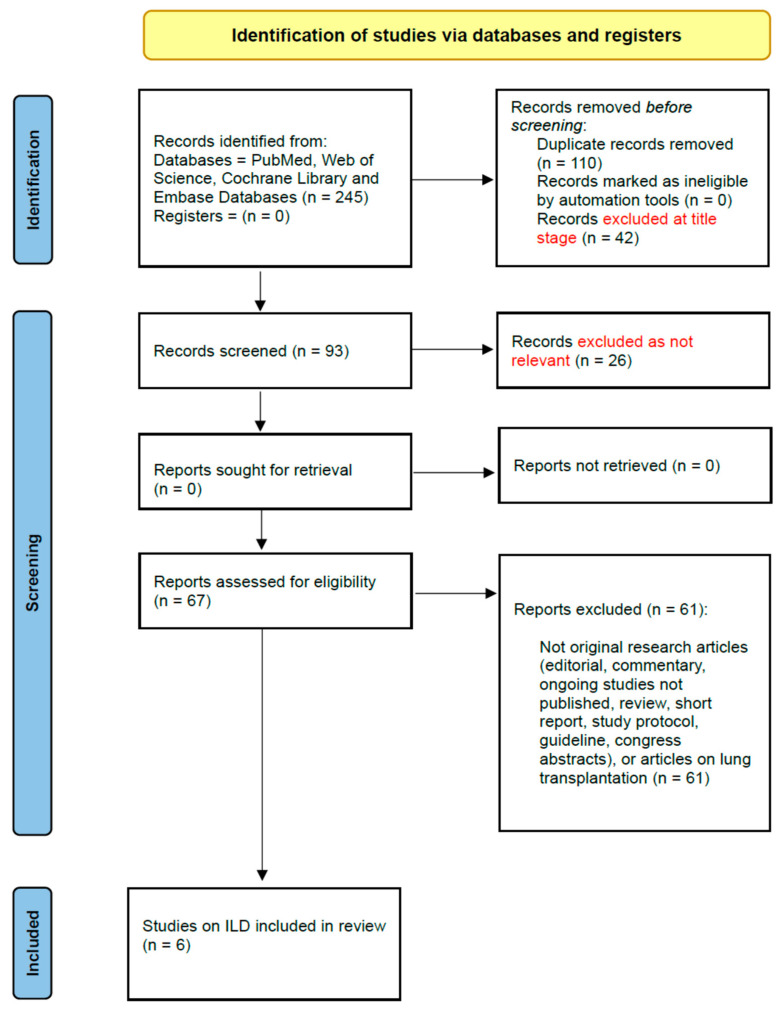
PRISMA Flowchart of the included studies on ILD. From: Page MJ, McKenzie JE, Bossuyt PM, Boutron I, Hoffmann TC, Mulrow CD et al. The PRISMA 2020 statement: an updated guideline for reporting systematic reviews. BMJ 2021;372:n71.

**Figure 2 geriatrics-08-00082-f002:**
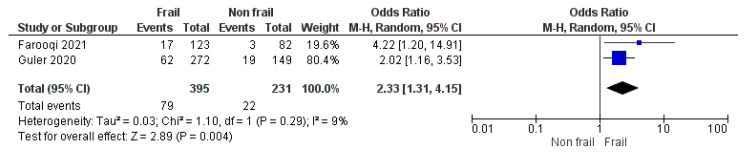
A meta-analysis of frailty and mortality in ILD. Forest plot describing effect of frailty on all-cause mortality. The mortality of ILD patients living with or without frailty were compared across two studies. Frailty vs. non-frailty and long-term mortality in ILD patients [23,24].

**Table 1 geriatrics-08-00082-t001:** Included studies on interstitial lung disease.

Author, Year	Country	Population Details/Study Design	Primary Outcomes	No. Patients	Age ^&^	Sex (% Male) (M:F)	Frailty Scale	Frailty Prevalence	Pre-Frailty Prevalence
Farooqi MAM, 2021 [21]	Canada	Outpatients with ILDs, 183 patients with IPF; prospective cohort study	Prevalence of physical frailty, impact of frailty on time to death and on time to lung transplantation in ILD patients	463	No age cut off 68 ± 11	55% 257 M 45% 206 F	**Fried Frailty Phenotype**	123 (26.5%)	258 (56%)
Guler SA, 2020 [22]	Canada	Outpatients with ILDs, 100 patients with IPF; prospective cohort study	Impact of frailty on time to death or lung transplantation, hospitalisations, time to hospital discharge, and health-related quality of life in ILD patients	540	No age cut off 67.4 ± 10.2	43% 232 M 57% 308 F	**Frailty Index** including 42 deficits (19 comorbidities and 23 items related to independence and self-care); frailty ≥ 0.21	272 (50.3%)	119 (22%)
Sheth JS, 2019 [24]	US	Outpatients with IPF; cross-sectional study	Prevalence of frailty and investigation of geriatric conditions and number of comorbidities in ILD patients	50	>65 yrs * 73.8 ± 5.4	66% 33 M 34% 17 F	**Fried Frailty Phenotype**	24 (48%)	20 (40%)
Guler SA, 2017 [25]	Canada	Outpatients with systemic sclerosis-associated ILD (SSc-ILD) and non-connective tissue disease fibrotic ILD (non-CT fibrotic ILD); cross-sectional study	Prevalence of frailty and comparison of the features of frailty between SSc-ILD and non-CT fibrotic ILD patients	253	No age cut off 60.5 ± 11.8 SSc-ILD 69.3 ± 9.9 Fibrotic ILD	45.4% 115 M 54.6% 138 F	**Frailty Index** including 42 deficits (19 comorbidities and 23 items related to independence and self-care); frailty ≥ 0.21	55% in SSc-ILD 50% in non-CT fibrotic ILD	21% in SSc-ILD 20% in non-CT fibrotic ILD
Milne KM, 2017 [26]	Canada	Outpatients with ILDs, 41 patients with IPF; cross-sectional study	Prevalence of frailty and investigation of predictors of frailty in ILD patients	129	>40 yrs * 69 ± 9.2	54.3% 70 M 45.7% 59 F	**Frailty Index** including 42 deficits (19 comorbidities and 23 items related to independence and self-care); frailty ≥ 0.21	64 (50%)	31 (24%)
Labrecque P-F, 2022 [23]	Canada	Outpatients with ILDs, 17 patients with IPF, and 15 control subjects; cross-sectional study	Prevalence of frailty and comparison with exercise capacity, functional mobility, muscle function and composition, and health-related quality of life between ILD patients and healthy subjects	36 patients 15 control subjects	No age cut off 70 ± 7 ILD patients 69 ± 7 Control subjects	78% 28 M 22% 8 F Control subjects 67% 10 M 33% 5 F	**Fried Frailty Phenotype**	9 (25%) in ILD patients	19 (53%) in ILD patients

* Only patients aged over 40 or over 65 enrolled in the study; ^&^ range unless stated as standard deviation (SD).

**Table 2 geriatrics-08-00082-t002:** Included studies on asthma.

Author	Country	Population Details/Study Design	Primary Outcomes	No. Patients	Age ^&^	Sex (% Male) (M:F)	Asthma Diagnosis/Definition Criteria	Frailty Scale	Frailty Prevalence	Pre-Frailty Prevalence
Landre’ B, 2020 [27]	France	Community-dwelling adults; prospective analysis with 26 years follow up	Prevalence of frailty in subjects with or without current asthma	12,345 Current asthma in 2015: 372 (3%)	69.8 ± 3.5	67% 250 M 33% 122 F	Self reported diagnosis	**Fried frailty phenotype**	13% (47/372)	38% (142/372)
Kusunose M, 2021 [28]	Japan	Outpatients with asthma in stable condition	Prevalence of frailty and influence of frailty on asthma severity	69	No age cut off 69.4 ± 10.8	49.3% 34 M 50.7% 35 F	GINA guidelines	**Kihon Checklist**	14.5% (10/69)	30.4% (21/69)
Hanlon P, 2018 [9]	UK	Community-dwelling adults; prospective analysis with 7 years follow up	Prevalence of frailty, all-cause mortality at 7-year follow-up and influence of frailty and pre-frailty on mortality	493,737 Asthma 57,169 (11.6%)	Age cut off 40–69 Range 37–73	46% 227,119 M 54% 266,618 F	Self reported diagnosis	**Fried frailty phenotype**	6% (3358/ 57,169)	42.1% (24,073/ 57,169)

^&^ Range unless stated as standard deviation (SD).

## Data Availability

All data sharing and collaboration requests should be directed to the corresponding author. The data underlying this article are available in the article and in its online Appendix A.

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
