# Peer review of "Frailty Prevalence and Association with Clinical Outcomes in Interstitial Lung Disease, Asthma, and Pleural Disease"

_geriatrics, 2023, doi:10.3390/geriatrics8040082_

Round 1

Reviewer 1 Report (Previous Reviewer 1)

See the comments in the attached file. 

Please check for some typos errors, especially in the tracked sentences. 

Author Response

REVIEWER 1

R1: I would ask the author to strenghten their choice to review specifically ILD, asthma, and pleural diseases, not considering, for example, other relevant conditions such as malignancies.

Re: We excluded lung cancer as an excellent systematic review and meta-analysis entitled “Frailty in patients with lung cancer” has been published in February 2022 on CHEST.

R1: Can we consider a follow up time between 17 and 36 months as “long term”? If authors say yes, please support this statement.

Re: We thank the reviewer for this important comment. We added the definition of “long term” to the Outcomes in the Methods section (page 3, lines 128-129).

R1: Please refers to the supplementaries in the main text.

Re: We thank the reviewer for this suggestion. We mentioned the S1_File Figure 1A, S1_Table 1B, S1_Table 1C in the text (Results, page 6, lines 197, 199, 209).

R1: A more in deep speculation over the observed differences in the prevalence of frailty between IPF and non-IPF ILDs should be carried out in the discussion.

Re: We thank the reviewer for raising this point. The studies included in the review described a wide range of ILDs (IPF and non-IPF), but the different prevalence of frailty seems to be more related to the “fibrosing phenotype”, of which IPF is considered the most typical. The presence of a “fibrosing” ILD, that is progressive, can worse the disease course and the time to mortality. However, there are limited data and need of further studies to explore this issue, including groups of patients with IPF and non-IPF. We could not explore the role of multimorbidity and other age-related health deficits, but future studies in ILDs might demonstrate that frailty is a further prognostic marker for risk stratification in these patients.

R1: Please clarify that these studies focused on an elderly population (clarify the mean age of the study groups).

Re: We thank the reviewer for this comment. The studies of Hanlon (2018) and Landrè (2020) were epidemiological. While Landrè included 372 subjects with current asthma and mean age of 69.8 years, Hanlon included 493737 individuals and 81% of those were aged < 65 years. Thus, we did not know the mean age of the subjects with asthma (n = 57169). Kusunose (2021) included outpatients with asthma with mean age of 69 years. We clarified these data in the Discussion (page 9, lines 292-294).

R1: Please highlight that the reviewed studies on asthma only provide a frailty phenotype and not frailty index and briefly discuss this.

Re: We thank the reviewer for this comment. The studies of Hanlon (2018) and Landrè (2020) were epidemiological and used the Fried Frailty Phenotype to assess frailty on a large scale. We cannot discuss the reasons of the methodology of these studies that probably did not use the Frailty Index for its complexity. The 5 criteria of Fried were adapted to be assessed by questionnaire in both studies.

R1: Considering the heterogeneity of the respiratory diseases that authors included in this review, I would comment the differences in the burden of frailty between asthma and ILD and the likely reasons for that.

Re: We thank the reviewer for this comment. We included only 3 studies on asthma of which 2 were epidemiological, and 6 studies on ILDs conducted in outpatients with confirmed diagnosis of ILD. Considering that the majority of patients with ILD have a severe disease with progressing dyspnea and reduced quality of life and only a few asthmatic patients have severe asthma with persistent shortness of breath and many exacerbations that influence daily activities, it is not surprising that frailty may have a higher degree in patients with ILDs.  

Reviewer 2 Report (Previous Reviewer 3)

I would thank the authors to have perfectly responding to almost of the comments.     
A misunderstanding persists regarding the abstract, and I have to insist regarding two of the main comments.

ABSTRACT

I have suggested that – in the abstract - the sentence “ The review found three studies relating to frailty in asthma, which did not examine its impact on outcomes in astham...” should move to the results. That was meaning that this sentence should move to the result section of the abstract.

 I would thank the authors to have also moving the sentence to the result section of the manuscript, that was a good decision [don't change anything].

 Regarding the abstract, I suggest to move:   The review found three studies that did not examine the impact of frailty on clinical outcomes in asthma. No studies relating to pleural disease and frailty were identified.” to the result section of the abstract as follows:

Results:  The review found three studies that did not examine the impact of frailty on clinical outcomes in asthma. No studies relating to pleural disease and frailty were identified. We identified 6 relevant studies incorporating…

The conclusion will end by the words “in this population.”

 MAIN COMMENT 1:

As already said, I was wondering why COPD was not included in the present work, and I asked the authors to justify the choice to group ILD, asthma and pleural disease without adding COPD.

The authors perfectly explained the reason: “Our team identified a number of respiratory disease areas where data on frailty were lacking. COPD was by far the largest. Therefore, we decided to pursue this area and we currently have this paper under review elsewhere. Further scoping searches identified Asthma, ILD and pleural as the next largest respiratory disease groups. Hence, we grouped our findings together for this paper.

They also said that they have revised the manuscript according to this comment, and that much of the Introduction and Discussion has been improved by a previous reviewer. However, I did not find any change related to the present comment in the introduction. I strongly suggest the authors to add their good justification in the introduction.

MAIN COMMENT 2/

The authors discussed the fact that among the studies on asthma, only one diagnose asthma according to established guidelines i.e. the GINA, and I had made a comment.

I agree with the response of the authors. But the fact is that the GINA guidelines are not the guidelines for epidemiological studies. Standardized and validated questionnaires are used to identified asthmatics in epidemiological studies. In the publication by Hanson, I don’t know (I did not find any information), but in the publication by Landré, in the discussion, it is stated that “it is interesting to note that 98% of the participants who reported having current asthma in 2002 also reported having symptoms of asthma or medication the same year, suggesting the reliability of annual asthma reports.

So, I strongly suggest to modulate the sentence (lines 291-294) by: Although all included studies were reliable, one study [30] assessed outpatients with diagnosis of asthma according to GINA guidelines, and the two other studies through questionnaires, meaning that the majority of participants had a mostly self-reported diagnosis.

Author Response

REVIEWER 2

R2: I would thank the authors to have perfectly responding to almost of the comments.     
A misunderstanding persists regarding the abstract, and I have to insist regarding two of the main comments.

ABSTRACT

I have suggested that – in the abstract - the sentence “ The review found three studies relating to frailty in asthma, which did not examine its impact on outcomes in asthma...” should move to the results. That was meaning that this sentence should move to the result section of the abstract.  I would thank the authors to have also moving the sentence to the result section of the manuscript, that was a good decision [don't change anything].

Regarding the abstract, I suggest to move:   “The review found three studies that did not examine the impact of frailty on clinical outcomes in asthma. No studies relating to pleural disease and frailty were identified.” to the result section of the abstract as follows:

Results:  The review found three studies that did not examine the impact of frailty on clinical outcomes in asthma. No studies relating to pleural disease and frailty were identified. We identified 6 relevant studies incorporating…

The conclusion will end by the words “in this population.”

Re: We thank the reviewer for allowing us to improve the abstract further. We modified the Abstract accordingly (page1, lines 24-25,30-33).

MAIN COMMENT 1:

As already said, I was wondering why COPD was not included in the present work, and I asked the authors to justify the choice to group ILD, asthma and pleural disease without adding COPD.

The authors perfectly explained the reason: “Our team identified a number of respiratory disease areas where data on frailty were lacking. COPD was by far the largest. Therefore, we decided to pursue this area and we currently have this paper under review elsewhere. Further scoping searches identified Asthma, ILD and pleural as the next largest respiratory disease groups. Hence, we grouped our findings together for this paper.

They also said that they have revised the manuscript according to this comment, and that much of the Introduction and Discussion has been improved by a previous reviewer. However, I did not find any change related to the present comment in the introduction. I strongly suggest the authors to add their good justification in the introduction.

Re: We apologise for that. We added the justification to the Introduction (page 2, lines 75-77).

MAIN COMMENT 2:

The authors discussed the fact that among the studies on asthma, only one diagnose asthma according to established guidelines i.e. the GINA, and I had made a comment.

I agree with the response of the authors. But the fact is that the GINA guidelines are not the guidelines for epidemiological studies. Standardized and validated questionnaires are used to identified asthmatics in epidemiological studies. In the publication by Hanlon, I don’t know (I did not find any information), but in the publication by Landré, in the discussion, it is stated that “it is interesting to note that 98% of the participants who reported having current asthma in 2002 also reported having symptoms of asthma or medication the same year, suggesting the reliability of annual asthma reports.

So, I strongly suggest to modulate the sentence (lines 291-294) by: Although all included studies were reliable, one study [30] assessed outpatients with diagnosis of asthma according to GINA guidelines, and the two other studies through questionnaires, meaning that the majority of participants had a mostly self-reported diagnosis.

Re: We thank the reviewer again as the change of the sentence makes the text balanced and clearer (Discussion, page 9, lines 297-300).

This manuscript is a resubmission of an earlier submission. The following is a list of the peer review reports and author responses from that submission.

Round 1

Reviewer 1 Report

I read with interest the paper by Verduri and Colleagues providing a comprehensive overview over frailty prevalence in subjects with selected respiratory conditions. 

I found the paper informative and focusing on a relevant topic, such as IPF. 

Here are my minor comments and suggestions aiming at improving the paper:

- Introduction: Frailty phenotype and frailty index (Rockwood's model or similar) express different biologic and clinical rationale, are suitable and useful in different settings, and therefore I would briefly clarify about these conceptual differences.

- Key Exposure of Frailty: When assessing for prevalence, I would stratify according to frailty detecting tool (frailty phenotype vs frailty index and see if significant differences in prevalence occur. It would be of interest to explore and analyze different performances within respiratory patient  

- Characteristics of Included Studies on asthma: Considering that age greatly vary in the pooled population (ranging between 37 and 80), were data somehow adjusted or stratified according to age? Furthermore, it could be intersting to highlight that early onset and late onset asthma dysplay different clinical trajectories over time and therefore could be differently associated with frailty. I would stress this concept in the discussion session

- Since Asthma and IPF are very different in their pathophysiology and natural history, I would discuss them separately by creating different sub-chapters in the discussion. 

Reviewer 2 Report

This review could be interesting but results are limited by a few studies reported, especially regarding  asthma and pleural disease. It represent a limit for scientific interest

minor revision to English language are required

Reviewer 3 Report

In the paper by Verduri et al., the authors conducted a review of observational studies to identify the prevalence of frailty in adult patients with interstitial lung diseases (ILD), asthma or pleural disease, and to assess the association between frailty and mortality, and morbidity (exacerbation and hospitalization). They also performed a meta-analysis.

The paper addresses a question of interest.  Indeed, frailty in relation with chronic respiratory diseases is often investigated disease by disease, and the present study investigates more than one.

However, the work suffers from many weaknesses, the main one being the lack of justification to include asthma and not COPD. Further, the strategy needs more explanation, and the authors failed to discuss in depth their results.

I already reviewed this paper in April for another journal. Therefore, I gave again similar comments and suggestions (see below).

 ABSTRACT

The authors should clearly state that the pooled analysis was done with studies on ILD rather than “when enough data allowed...”.
The sentence that “… found three studies relating to frailty in asthma, which did not examine its impact on outcomes” should move to the results.
Conclusion should not only summarize the results. 

INTRODUCTION

As already said, I am wondering why COPD was not included in the present work. If asthma was, why not COPD? Literature on Frailty and COPD is not so extensive. The answer is probably at this end of the paper (lines 240-242) when the authors said that “Similarly, other lung diseases such as COPD showed an increased risk of mortality in patients with COPD [Verduri et al. Manuscript in review].” Is this paper accepted or published now?

I ask the authors to justify the choice to group ILD, asthma and pleural disease without adding COPD.

 METHOD

Search strategy and Study selection:
Regarding ILD: The search strategy seems not to be the same in the text (Lines 78-93) and in the figures or Supplementary Tables or Figure. 245 records were identified in Figure 1 (135 when excluding the duplicates), 118 identified in S1 Table 3. Reasons of exclusions in the Prisma flow-chart for ILD should be given (for n=42, n=26, and n=61).
Regarding asthma: The other reasons (n=39) in the Prisma flow-chart (S1 Figure 1A) should be detailed. Why the search strategy for asthma is not given?
The Prisma flow-chart for pleural disease (S1 Figure 1B) is uninformative (n=0) and should be deleted.

This part of the paper should be clearly detailed and informative.

Inclusion criteria (lines 96-97 and 115-117)
The authors said that one of the inclusion criteria is having prevalence of Frailty and related-morbidity and mortality as outcomes. Then they said that prevalence of frailty was the primary one and that mortality, exacerbation rate or number of hospitalizations were the secondary ones.

Among the studies described in Tables 1 and 2, it seems that some of them did not investigate one of the secondary outcomes. The sentence in the abstract also added confusion “… found three studies relating to frailty in asthma, which did not examine its impact on outcomes”.
I suggest to add a column describing the secondary outcome in Tables 1 and 2. If needed, I suggest to rephrase the sentence lines 96-97.

Which statistical software was used? (line 134).

RESULT

Line 148: the authors said that 135 records were identified. Where is this number indicate in Flow-chart? I suppose that from Figure 1 we have to calculate 245-110=135. I strongly suggest the authors to present clear and self-explicit PRISMA flow-chart for ILD and asthma.

Thanks to having added the min and max prevalence of Frailty.

S1 Table 2A: what does * mean?

 DISCUSSION

The discussion should “discuss” the results more in depth. The parts which summarize the results or repeat some of them should be shortened (lines 225-230). The summary of the results on asthma (Lines 274-277) is not discussed.

 Note that the word “demonstrated” should be avoided along the text and in the supplementary file (title of S1 Figure 3). Observational studies do not demonstrate nothing.

 In the discussion, there is a reference to a paper from the same first author (Verduri) on COPD that may highlight the reason why COPD is not in the present review (see my previous comment on the need of a clarification and a scientific justification in the introduction to understand why asthma and not COPD). If this paper is referenced, other papers on COPD should also be referenced and discussed, e.g. the recent paper by Xu J, Xu W, Qiu Y, Gong D, Man C, Fan Y. Association of Prefrailty and Frailty With All-Cause Mortality, Acute Exacerbation, and Hospitalization in Patients With Chronic Obstructive Pulmonary Disease: A Meta-Analysis. J Am Med Dir Assoc. 2023 May 5:S1525-8610(23)00324-9. doi: 10.1016/j.jamda.2023.03.032. Epub ahead of print. PMID: 37150209. Or the paper by

I also suggest to discuss the lack of data regarding Frailty and ACOS (asthma-COPD overlap syndrome) especially in a journal as Geriatrics.

 The authors discussed the fact that among the studies on asthma, only one diagnose asthma according to established guidelines i.e. the GINA in 69 patients. The diagnose of asthma in large epidemiological studies among community-dwelling adults (here 12345 and more than 57000) is not the same as the diagnose of asthma in small sample size of clinical studies. Standardized and validated definitions of asthma based on questionnaires are adequate to study this disease in epidemiology. Based on this comment, I am wondering if the criteria of the quality assessment of the studies are good regarding epidemiological studies on asthma. The authors should consider this point, and think to re-consider their criteria regarding asthma in S1Table 2A or discuss this point.

 The authors lacked to discuss the interrelation between frailty and respiratory disease on mortality, and would have gain to discuss their results also according to the type of observational studies they have selected: cross-sectional, longitudinal…

 The papers by Onder et al. and that by Angulo et al. (see below) should be discussed or added in the introduction.

Onder G, Vetrano DL, Marengoni A, Bell JS, Johnell K, Palmer K; Optimising Pharmacotherapy through Pharmacoepidemiology Network (OPPEN). Accounting for frailty when treating chronic diseases. Eur J Intern Med. 2018 Oct;56:49-52. doi: 10.1016/j.ejim.2018.02.021. Epub 2018 Mar 9. PMID: 29526651.

Angulo J, El Assar M, Rodríguez-Mañas L. Frailty and sarcopenia as the basis for the phenotypic manifestation of chronic diseases in older adults. Mol Aspects Med. 2016 Aug;50:1-32. doi: 10.1016/j.mam.2016.06.001. Epub 2016 Jun 28. PMID: 27370407.